

# The inflated mitochondrial genomes of siphonous green algae reflect processes driving expansion of noncoding DNA and proliferation of introns

Sonja I. Repetti[1], Christopher J. Jackson[1], Louise M. Judd[2],
Ryan R. Wick[2], Kathryn E. Holt[2] and Heroen Verbruggen[1]

[1] School of BioSciences, University of Melbourne, Melbourne, VIC, Australia
[2] Department of Infectious Diseases, Monash University, Melbourne, VIC, Australia

Corresponding author
Sonja I. Repetti,
sonjarepetti@live.com.au

## ABSTRACT

Within the siphonous green algal order Bryopsidales, the size and gene arrangement of chloroplast genomes has been examined extensively, while mitochondrial genomes have been mostly overlooked. The recently published mitochondrial genome of *Caulerpa lentillifera* is large with expanded noncoding DNA, but it remains unclear if this is characteristic of the entire order. Our study aims to evaluate the evolutionary forces shaping organelle genome dynamics in the Bryopsidales based on the *C. lentillifera* and *Ostreobium quekettii* mitochondrial genomes. In this study, the mitochondrial genome of *O. quekettii* was characterised using a combination of long and short read sequencing, and bioinformatic tools for annotation and sequence analyses. We compared the mitochondrial and chloroplast genomes of *O. quekettii* and *C. lentillifera* to examine hypotheses related to genome evolution. The *O. quekettii* mitochondrial genome is the largest green algal mitochondrial genome sequenced (241,739 bp), considerably larger than its chloroplast genome. As with the mtDNA of *C. lentillifera*, most of this excess size is from the expansion of intergenic DNA and proliferation of introns. Inflated mitochondrial genomes in the Bryopsidales suggest effective population size, recombination and/or mutation rate, influenced by nuclear-encoded proteins, differ between the genomes of mitochondria and chloroplasts, reducing the strength of selection to influence evolution of their mitochondrial genomes.

## INTRODUCTION

Although they share many unifying features, the green algae (Chlorophyta) are a diverse group with genomes that vary considerably in structure and gene content (*Yurina & Odintsova, 2016*). Most work on Chlorophyta genomes to date has focused on organellar genomes, with in excess of 150 chloroplast and 70 mitochondrial genomes published on GenBank. These organelle genome data have been used for phylogenetic analyses to resolve lineage relationships within the Chlorophyta (*Fučíková et al., 2014*; *Cremen et al., 2018*) and

to investigate organellar genome structural adaptation and unique features. As with many plastid-bearing taxa, more effort has been made to sequence and characterise chloroplast genomes rather than mitochondrial, as plastid genomes tend to be favoured for their use in phylogenetic studies due to high coding content, relatively conserved genome and simple inheritance patterns (*Smith & Keeling, 2015*; *Fang et al., 2017*). There are some notable exceptions in the Chlorophyta such as the fragmented hairpin plasmids seen in Cladophorales plastids (*Del Cortona et al., 2017*) and the large and highly repetitive *Acetabularia acetabulum* (Dasycladales) plastid genome (*De Vries et al., 2013*). Despite this, mitochondrial genomes tend to show greater abnormalities in structure than plastid genomes (*Burger, Gray & Lang, 2003*), and some of the most unusual genomic features have been observed in eukaryotic mitochondrial genomes (*Smith & Keeling, 2015*).

Comparative studies in green algal chloroplast and mitochondrial genomes have found that many instances of genome size variation are not due to gene gain or loss, but rather reduction or expansion of noncoding DNA including introns (*Burger, Gray & Lang, 2003*; *Smith et al., 2013*; *Marcelino et al., 2016*). In the Chlamydomonadales, *Smith et al. (2013)* observed an increase in chloroplast DNA from approximately 60% noncoding DNA in smaller-sized unicellular lineages to greater than 80% in multicellular *Volvox carteri*, while the number of genes differs by only two between the smallest and largest genomes. A similar pattern was observed for their mitochondrial DNA (*Smith et al., 2013*). According to *Smith & Lee (2010)*, the large noncoding DNA content in *V. carteri* genomes could support the mutational-hazard hypothesis (MHH). The MHH proposes that excess DNA is more likely to accumulate in genomes with a low mutation rate and small effective population size ($N_e$) (*Lynch, 2006*; *Smith, 2016*) due to noncoding DNA imposing a fitness burden by increasing the potential for harmful mutations. Such excess DNA can be eradicated more effectively when there is a larger effective population size for natural selection to act upon through purifying selection (*Lynch & Walsh, 2007*). The MHH might explain the streamlining of organelle genomes within various lineages including prasinophytes, red algae, and some fungi, which have high estimated mutation rates (*Smith, 2016*). A large reduction in $N_e$ is expected for multicellular organisms compared with unicellular, therefore purifying selection should be less efficient meaning there is less restriction on the accumulation of noncoding DNA. This may explain the difference in noncoding content between unicellular members of the Chlamydomonadales, such as *Chlamydomonas*, and colonial *V. carteri* (*Smith et al., 2010*). Various studies have supported and challenged the MHH (*Smith, 2016*), but on the whole emphasise the importance of mutation and genetic drift in shaping organelle genomes (*Lynch, 2006*; *Smith, 2016*).

It has also been suggested that selection can influence the mutation rate of genomes. The drift-barrier hypothesis (*Lynch et al., 2016*) proposes that selection acts to reduce the mutation rate with an overall limit set by genetic drift. In genomes with very high mutation rates, 'antimutators' such as DNA repair proteins are advantageous and decrease the mutation rate until the strength of selection is matched by that of genetic drift and mutation bias (*Lynch et al., 2016*). In contrast, in genomes with low mutation rates, antimutators will not be advantageous enough to be selected for, whilst having mild

mutators will not be sufficiently disadvantageous to be selected against. Therefore, mutation rate will increase until selection is strong enough to prevent the genome evolving a higher mutation rate (*Lynch et al., 2016*). *Krasovec et al. (2017)* combined a review of literature with mutation rate estimates for four prasinophytes and found support for this theory, with mutation rate tending to decrease as $N_e$, therefore strength of selection, increases.

The effective population size of genomes influences the efficacy of selection relative to genetic drift (*Ness et al., 2016*). Within genomes, loci linked to sites under selection will exhibit locally reduced $N_e$ due to hitchhiking, where mutations that are not necessarily optimal are fixed because they are located close to another gene undergoing selection (*Smith & Haigh, 1974*), background selection, when selection acts against linked deleterious alleles that are retained in a population by mutation (*Charlesworth, 1994*), and Hill–Robertson (HR) interference (*Platt, Weber & Liberles, 2018* and references therein). Under the HR effect (*Hill & Robertson, 1966*), weak selection acting on linked sites reduces the overall effectiveness of selection due to linkage disequilibrium (*Comeron, Williford & Kliman, 2008*). Recombination breaks up such linkage and prevents local reduction in $N_e$ due to selection acting upon linked sites (*Cutter & Payseur, 2013*; *Platt, Weber & Liberles, 2018*). It has been observed that genes located in regions of low recombination have a greater number of introns, and these introns have greater lengths (*Comeron & Kreitman, 2000*). It has been proposed that introns may be advantageous in reducing intragenic HR effect and are hence retained, as they might increase rates of recombination that lower intragenic HR effects (*Comeron, Williford & Kliman, 2008*). *Comeron & Kreitman (2000)* suggest that longer introns are advantageous in regions of low recombination, as they reduce the HR effect relative to a shorter intron by increasing the recombination rate between mutations in different exons, and that they will be maintained due to hitchhiking effects between favourable mutations and the longest intron variant.

The class Ulvophyceae within the Chlorophyta show high morphological diversity, including a range of cell configurations (*Cocquyt et al., 2010*; *Fang et al., 2017*), and are thus a useful model for examining the evolution of genome features under differing evolutionary constraints. The order Bryopsidales are a lineage of siphonous seaweeds with thali comprised of a single giant tubular cell containing cytoplasm with a large number of nuclei and other organelles free to move around the entire plant (*Vroom & Smith, 2001*, *2003*; *Verbruggen et al., 2009*; *Mine, Sekida & Okuda, 2015*). Within the Bryopsidales, the size and gene arrangement of chloroplast genomes has been found to vary considerably while gene content was much less variable (*Cremen et al., 2018*). While appreciable short-read sequencing data have been generated for the Bryopsidales, mitochondrial genomes have not been found to assemble well from the sequencing libraries used to assemble chloroplast genomes, and the first bryopsidalean mitochondrial genome was published only very recently for the sea grape *Caulerpa lentillifera* (*Zheng et al., 2018*). This 209,034 bp circular-mapping genome is the largest Chlorophyta mitochondrial genome published to date, an order of magnitude larger than the average for Chlorophyta mitochondrial genomes and rich in intergenic sequences and introns, the causes of which have not been examined.

Unlike the genome expansion observed in the Chlamydomonadales, which appears to occur in both organellar compartments (*Smith et al., 2013*; *Zhang et al., 2019*), *C. lentillifera*'s mitochondrial genome is much larger than its chloroplast genome. This is unusual for green algae where mitochondrial genomes are generally smaller than chloroplast genomes (*Leliaert et al., 2012*). Furthermore, mitochondrial genome expansion is less frequent in green algae compared with plants (*Mower, Sloan & Alverson, 2012*). While *Zheng et al. (2018)* did not propose an evolutionary explanation for these findings, the inflated noncoding regions described are consistent with previous studies in plants showing that the variation in the sizes of their mitochondrial genomes is predominantly explained by changes in noncoding DNA content including repeats, introns, intergenic DNA and DNA of foreign origin (*Yurina & Odintsova, 2016*). Therefore, the mitochondrial genome of *C. lentillifera* provides an interesting case in the Chlorophyta to examine the evolution of organellar genomes and, in particular, the processes driving genome expansion. However, it remains undetermined whether a large mitochondrial genome is a feature only of *Caulerpa*, or if it is in fact characteristic of the order Bryopsidales.

In this paper, we focus on another member of the Bryopsidales that is currently of considerable ecological interest: the genus *Ostreobium*, an endolithic limestone-boring alga (*Verbruggen & Tribollet, 2011*). *Ostreobium* is found in a diverse range of calcium carbonate environments around the world, is abundant in the skeleton of many coral species, and it is one of the most common genera of endolithic autotrophs in coral reefs (*Tribollet, 2008*). Endolithic algae like *Ostreobium* have a number of observed and predicted roles in the coral skeleton (*Ricci et al., 2019*). While their boring can destabilise coral skeletons and impact on reef structure (*Schlichter, Kampmann & Conrady, 1997*), they may have a mutualistic relationship with corals, providing them with metabolites that could enable them to survive bleaching events (*Schlichter, Kampmann & Conrady, 1997*; *Tribollet, 2008*). Its endolithic lifestyle means that *Ostreobium* inhabits low light habitats with limited available photosynthetically active radiation (*Wilhelm & Jakob, 2006*; *Magnusson, Fine & Kühl, 2007*) and it has been shown to have far red-shifted absorption spectra, allowing it to use the long wavelengths of light available in the coral skeleton (*Wilhelm & Jakob, 2006*; *Magnusson, Fine & Kühl, 2007*; *Tribollet, 2008*).

*Ostreobium* species have consistently small chloroplast genomes relative to the median for Bryopsidales of 105 kb (*Cremen et al., 2018*), with the chloroplast genome of *Ostreobium* strain *HV05042* the most compact found so far (80,584 bp) in the Ulvophyceae (*Marcelino et al., 2016*; *Verbruggen et al., 2017*). *Marcelino et al. (2016)* identified only three introns in the *Ostreobium quekettii* chloroplast genome, and there was an overall reduction in intergenic regions that resulted in its reduced size. They hypothesised that this might reflect evolution in response to resource constraints in its low light environment (*Marcelino et al., 2016*), but did not examine the mitochondrial genome for evidence of similar selection pressures.

The goal of this study is to evaluate the characteristics of mitochondrial genomes in the Bryopsidales, as well as to determine the evolutionary forces that have shaped organelle genome dynamics in the Bryopsidales. Our approach consists of the sequencing, assembly

and annotation of the *O. quekettii* mitochondrial genome, as well as analysis of noncoding elements: repeats, introns and intron-encoded open reading frames (ORFs). We also compare the *O. quekettii* mitochondrial genome with that of *C. lentillifera*, interpreting results in the context of existing evolutionary genomics theory.

## MATERIALS AND METHODS

### Culturing and DNA extraction

*Ostreobium quekettii* (SAG culture collection strain 6.99) was cultured in F/2 media on a 14H–8H light/dark cycle at ~19 °C. Total DNA was extracted using a modified cetyltrimethylammonium bromide method described in *Cremen et al. (2016)*.

### DNA sequencing

Total DNA was sequenced using Illumina sequencing technology (HiSeq 2000, 150 bp paired-end reads, ~40 GB data), at Novogene, Beijing. Reads were trimmed with Cutadapt v1.12 (*Martin, 2011*) using the parameters −e 0.1 −q 10 −O 1 −a AGATCGGAAGAGC.

Total DNA was also extracted as above and sequenced using nanopore sequencing technology (MinION; Oxford Nanopore Technologies, Oxford, UK), producing 1,889,814 reads and ~9.5 GB data. Reads were filtered using Filtlong (https://github.com/rrwick/Filtlong), removing reads less than 1,000 bp in length, with average quality of less than 60 (as defined by the Phred scores), or minimum quality over a sliding window of less than 40.

### Genome assembly

The Illumina data were assembled using Spades v3.12.0 (*Bankevich et al., 2012*) with the—careful option. The Nanopore data were assembled using Canu 1.7.1 (*Koren et al., 2017*) with the parameters genomeSize = 300 m corOutCoverage = 10,000 corMhapSensitivity = high corMinCoverage = 0.

A hybrid (combined long-read and short-read) genome assembly was performed with MaSuRCA v3.2.7 (*Zimin et al., 2013*), using the Illumina reads as short-read input. For long-read input, contigs produced by Spades were treated as 'pseudo–Nanopore' reads and combined with Nanopore data, and this combined dataset was used.

### Transcriptome

Total RNA was extracted using Plant RNA reagent (Thermofisher, Waltham, MA, USA). A strand-specific 100 bp paired-end library was constructed and sequenced using Illumina HiSeq 2500 (ENA study accession number PRJEB35267). Quality filtering of reads was performed using Trimmomatic 0.39 (*Bolger, Lohse & Usadel, 2014*) with the following settings: LEADING:3 TRAILING:3 SLIDINGWINDOW:4:20. A transcriptome was constructed using Trinity version 2.8.3 (*Grabherr et al., 2011*).

### Identification and manual curation of mitochondrial genome

The **C. lentillifera** mitochondrial genome (GenBank accession KX761577.1) (*Zheng et al., 2018*) was used as the query in a BLASTn search against the long-read Canu assembly within Geneious version 11.1.2 (*Kearse et al., 2012*) with default settings. Only a single contig was identified as a likely candidate for the *O. quekettii* mitochondrial genome based

on the results of BLASTx searches of sections of this candidate contig against the NCBI nr and nt databases.

This contig was used as the query in a BLASTn search against the *O. quekettii* Spades short-read assembly, within Geneious with default settings. Top hits were aligned with the long-read contig within Geneious. Scaffolds from the short-read assembly were used as a reference to manually correct the long-read contig within Geneious. In order to verify that the genome was circular-mapping, we searched for a short-read scaffold spanning across both ends of the long-read contig when aligned.

This manual result was compared with the mitochondrial contig identified, through blast searches against the NCBI nr and nt databases, in the MaSuRCA hybrid assembly by alignment in Geneious.

## Genome annotation

The genome was initially annotated using MFannot (*Beck & Lang, 2010*), and DOGMA (*Wyman, Jansen & Boore, 2004*) with very relaxed settings (protein identity cut off 25%, RNA identity cut off 30%). Annotations of predicted protein coding genes were confirmed through extraction of ORFs and BLAST searches of these against the NCBI nr and nt databases, as well as alignment with transcripts that were recovered as hits from BLASTn searches against the *O. quekettii* transcriptome within Geneious, using default settings.

tRNAs were identified using tRNAscan-SE (*Lowe & Chan, 2016*), tRNAfinder (*Kinouchi & Kurokawa, 2006*), ARAGORN (*Laslett & Canback, 2004*) and tRNADB-CE's BLASTN/Pattern Search (*Abe et al., 2010*). rRNAs were identified with RNAmmer (*Lagesen et al., 2007*) and RNAweasel (*Lang, Laforest & Burger, 2007*).

Repeats were identified using the tandem repeats database (*Gelfand, Rodriguez & Benson, 2007*), the RepeatFinder package in Geneious with a minimum repeat length of 50 bp (*Pombert et al., 2004*) and REPuter (*Kurtz et al., 2001*) with minimal repeat size setting of 12 bp (*Smith & Lee, 2009*). Forward, reverse, complement, and reverse complement repeats were all considered under REPuter.

A map of the genome was created with Circos (*Krzywinski et al., 2009*) and manually annotated in Inkscape 0.92 (www.inkscape.org). The sequence for the *O. quekettii* mitochondrial genome is available on GenBank (Accession number MN514984).

## Open reading frames

Open reading frames were predicted using ORF finder in Geneious with a minimum length setting of 300 bp. These were used as queries in BLASTx searches against the NCBI nr and nt databases ($e$ value = $e^{-1}$), and the translated ORFs were used as queries in a batch sequence search against the Pfam database (*Finn et al., 2017*). Only ORFs that had BLAST results and identified Pfam domains were retained in the final genome annotation. In order to determine relationships between ORFs indicative of common origin, the annotated ORFs along with those from the chloroplast genome of *O. quekettii* (*Marcelino et al., 2016*) and the mitochondrial genome of *C. lentillifera* (*Zheng et al., 2018*) were clustered based on all-against-all BLAST+ similarities using CLANs (*Frickey & Lupas, 2004*). This was performed through the MPI Bioinformatics toolkit (*Zimmermann et al., 2018*),

with the BLOSUM62 scoring matrix and extraction of BLAST HSPs up to *e*-values of 1e−4. The output from CLANs was annotated in Inkscape.

### Introns

Intron boundaries were inferred by aligning predicted genes with their corresponding transcripts and intron-lacking homologues from other green algae. Intron class was predicted using RNAweasel, and Rfam sequence search (*Griffiths-Jones et al., 2003*).

In order to identify potential common origins of introns in the *O. quekettii* and *C. lentillifera* mitochondrial genomes as well as the *O. quekettii* plastid genome, we used a distance-based clustering technique. In contrast to the analysis of ORFs above, this analysis used DNA sequences and used introns that were not interrupted by ORFs, in the hope that this would represent a subset of the data that could be more easily aligned if they contained conserved regions. From the intronic DNA sequences, a distance matrix was constructed by comparing the introns from these genomes using Clustal Omega (*Sievers et al., 2011*) with the '—distmat-out' and '—full' flags. This distance matrix was used as the input to construct a neighbour joining (NJ) tree using Neighbor within the PHYLIP package (*Felsenstein, 2005*). We also constructed a distance matrix using a subset of these introns from only the *O. quekettii* chloroplast and mitochondrial genomes and this was used to construct a NJ tree. This NJ tree was further visualised and annotated in MEGA (*Kumar et al., 2018*). Clusters of introns identified in the NJ tree were aligned in Geneious using MAFFT to better evaluate similarity. Insertion sites of introns were also compared between the O. quekettii and *C. lentillifera* genomes by aligning gene sequences in Geneious using MAFFT.

### Rates of evolution

To obtain an estimate of the relative rates of evolution in the *O. quekettii* mitochondrial and chloroplast genomes, we aligned all the protein coding and rRNA genes common between the *O. quekettii* and *C. lentillifera* mitochondrial and chloroplast genomes in Geneious using the default aligner. We generated estimates of base substitutions per site between sequences using the Jukes–Cantor model (*Jukes & Cantor, 1969*) in MEGA. We also attempted to obtain $d_N/d_S$ ratios for gene alignments using PAL2NAL (*Suyama, Torrents & Bork, 2006*).

### Mitochondrion-targeted nuclear genes

In the context of evaluating hypotheses around mitochondrial DNA repair machinery (see below), we searched for homologues of nuclear-encoded DNA repair proteins that are targeted to the mitochondrion in higher plants: MSH1, RECA proteins and OSB1. *Arabidopsis* sequences were used as queries in BLAST searches with default settings against the nuclear contigs in the *O. quekettii* hybrid assembly, as well as the *O. quekettii* transcriptome in Geneious. We searched for putative targeting signals to the mitochondrion with DeepLoc-1.0 using default settings (*Almagro Armenteros et al., 2017*).

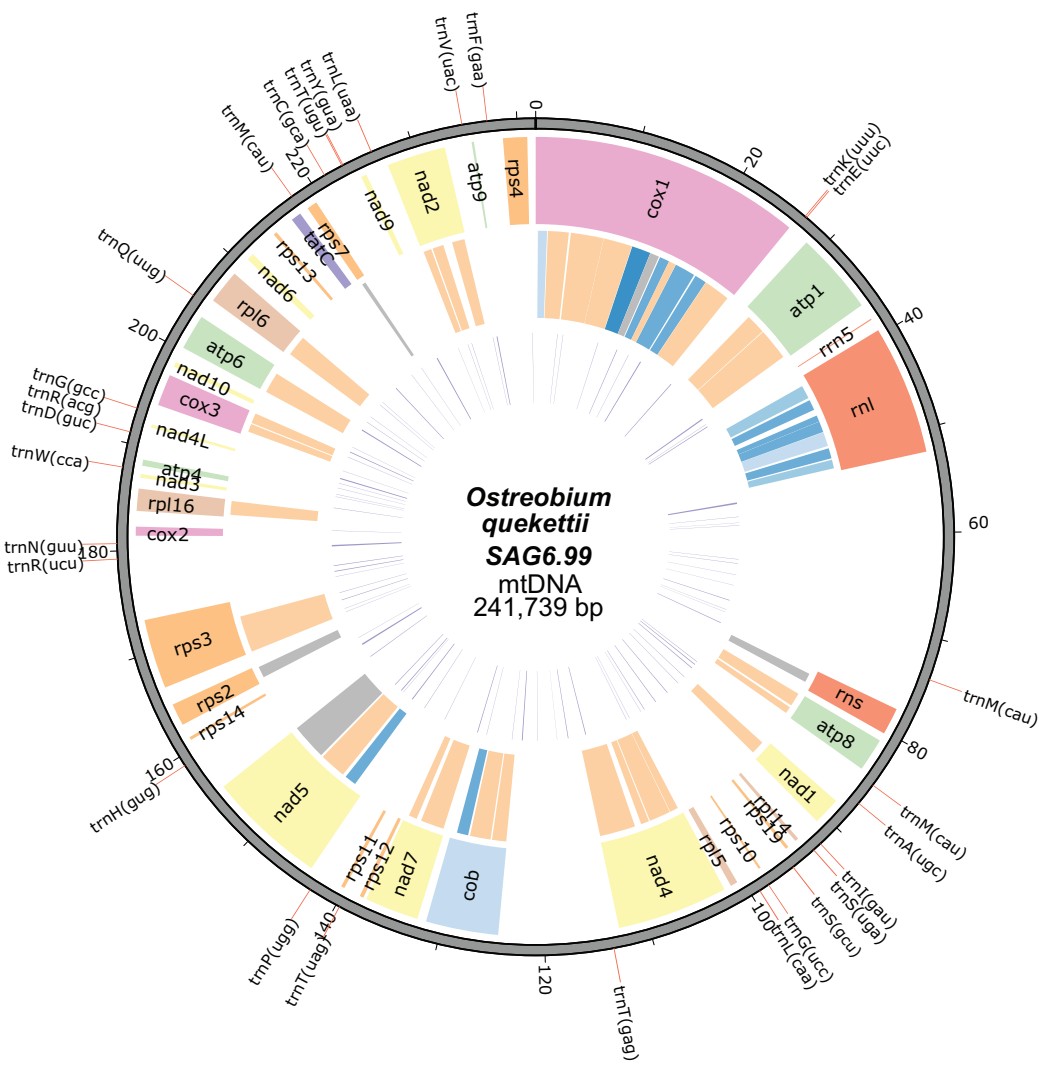

**Figure 1 Mitochondrial genome map of _Ostreobium quekettii_ SAG6.99.** The position of tRNAs are shown on the outer track (red lines). The first inner circle represents the position, size and the names of the protein-coding and rRNA genes. The introns are shown in the second inner circle and are colour coded according to intron types/subtypes: group I derived (very light blue), group IA (light blue), group IB (blue), group ID (dark blue), group II (orange), unknown (grey). The third inner circle represents the position and length (line thickness) of repeats.               

## RESULTS

### Ostreobium mitochondrial genome

The mitochondrial genome was captured by a single contig in the Canu assembly, but six separate contigs in the Spades assembly. The genome was captured by a single contig in the hybrid genome assembly that showed 99.97% pairwise identity with our manually corrected version of the genome. The combination of long- and short-read data improved this assembly and allowed us to capture the entire genome with a high level of accuracy.

The mitochondrial genome of _O. quekettii_ SAG6.99 assembled into a 241,739 bp circular-mapping molecule (Fig. 1). All 64 codons are used (Table S1) and the 28 tRNAs

**Table 1 Summary of coding and noncoding content of the mtDNA of *Ostreobium quekettii* SAG6.99.**

|  | Percent of total noncoding DNA | Percent of overall genome |
|---|---|---|
| Coding (rRNA, tRNA, ORFs, protein coding genes) |  | 25% |
| Repeats | 7% | 5% |
| Intergenic DNA | 61% | 46% |
| Introns (including ORFs) |  | 39% |
| Introns (excluding ORFs) | 39% | 29% |
| Total noncoding DNA (excluding ORFs) |  | 75% |
| Total intronic and intergenic DNA (including ORFs) |  | 85% |

**Table 2 Genes, introns and open reading frames present in mtDNA of *Ostreobium quekettii* SAG6.99. For further information on introns and ORFs, see Supplemental Material.**

|  | Number in genome |
|---|---|
| Protein coding genes | 34 |
| rRNA | 3 |
| tRNA | 28 |
| Genes containing introns | 18 |
| Introns | 47 |
|     Group I | 14 |
|     Group II | 28 |
|     Unclear | 5 |
| Introns containing ORFs | 18 |
| ORFs | 20 |
|     Intronic | 20 |
|     Intergenic | 0 |

encoded by the *O. quekettii* mtDNA (Table S2) appear to be sufficient to recognise all of these codons assuming the standard genetic code and maximum use of wobbling and superwobbling (*Alkatib et al., 2012*). Only 25% of the genome is coding DNA (Table 1). The genome encodes 3 rRNAs and 28 tRNAs (Tables 2 and 3), resembling other green algal mitochondrial genomes (Table S3). It also encodes 34 named protein-coding genes commonly found in green algae (Table 3; Table S3). This does include some genes that are less common in Chlorophyta mitochondrial genomes: *nad*10, which is absent from the mtDNA of sequenced land plants and many green algae (*Mower, Sloan & Alverson, 2012*), and *tatC*, a gene encoding a component of the inner membrane TAT translocase responsible for transporting folded proteins across the membrane in bacteria but whose function in mitochondria remains unclear (*Carrie, Weißenberger & Soll, 2016*; *Petrů et al., 2018*).

The overall GC content of the genome is 48.3% (Table 1), which is higher than the average for eukaryotic mitochondrial genomes (38%) (*Smith & Lee, 2008*) but similar to that of *C. lentillifera* (50.9%) (*Zheng et al., 2018*) and not extreme for green algae

**Table 3 Protein coding and ribosomal genes present in the mtDNA of *Ostreobium quekettii* SAG6.99.**

| Protein genes | |
|---|---|
| Complex I (*nad*) | *nad*1, *nad*2, *nad*3, *nad*4, *nad*4L, *nad*5, *nad*6, *nad*7, *nad*9, *nad*10 |
| Complex III (*cob*) | *cob* |
| Complex IV (*cox*) | *cox*1, *cox*2, *cox*3 |
| Complex V (*atp*) | *atp*1, *atp*4, *atp*6, *atp*8, *atp*9 |
| SSU ribosomal proteins (*rps*) | *rps*2, *rps*3, *rps*4, *rps*7, *rps*10, *rps*11, *rps*12, *rps*13, *rps*14, *rps*19 |
| LSU ribosomal proteins (*rpl*) | *rpl*5, *rpl*6, *rpl*14, *rpl*16 |
| Ribosomal RNAs | *rrn*5, *rrn*S, *rrn*L |
| Putative Protein Transporter | *tatC* |

(*Del Vasto et al., 2015*). It does however vastly exceed the GC content of the *O. quekettii* chloroplast genome (31.9%) (*Marcelino et al., 2016*).

The *O. quekettii* mitochondrial genome contains 373 repeats, found in both introns and intergenic DNA, that represent 5% of the total genome (Fig. 1; Table 1), with a minimum length of 31 and maximum of 299 bp (mean 107.3 ± 61.7 SD). This is similar to the mitochondrial genome of *C. lentillifera*, where we estimated 349 repeats represent approximately 4% of the genome, with a minimum length of 25 bp and maximum of 163 bp (mean 68.3 ± 30.1 SD).

## Introns

Most of the genome's inflated size (75%) is from expanded intergenic and intronic regions rather than extra coding material. Eighteen of the 34 named protein-coding genes contain one or multiple intron(s) (Table 2; Table S4). Introns include both group I and group II introns, as well as five whose class could not be confidently determined (Table 2; Table S4). The *Ostreobium* mitochondrial genome contains 47 introns, compared with only 29 in the mtDNA of *C. lentillifera* (*Zheng et al., 2018*). Group II introns, which appear to have been abundant early in the evolution of Bryopsidales plastid genomes (*Cremen et al., 2018*) and are the dominant type in plant mtDNA but less common elsewhere (*Lang, Laforest & Burger, 2007*), are the dominant type in the *O. quekettii* mtDNA. This contrasts with the *C. lentillifera* mitochondrial genome, which also contains both types of introns but with group I introns predominating (*Zheng et al., 2018*).

Open reading frames-lacking introns from the *O. quekettii* and *C. lentillifera* mitochondrial genomes mostly form separate clusters in NJ trees of pairwise DNA distances (Fig. S1). Alignments of the few introns that do cluster together did not show convincing homology between *C. lentillifera* and *O. quekettii* mitochondrial genomes (alignments not shown), nor between introns in the *O. quekettii* mitochondrial and plastid genomes (Fig. S2). Furthermore, comparing insertion sites for introns between *O. quekettii* and *C. lentillifera* only revealed five sites where intron insertion sites of the same type of intron were within 3 bp of each other, and only a single site where they matched exactly (Table S4). The NJ tree did identify groups of similar group II introns within the *O. quekettii*

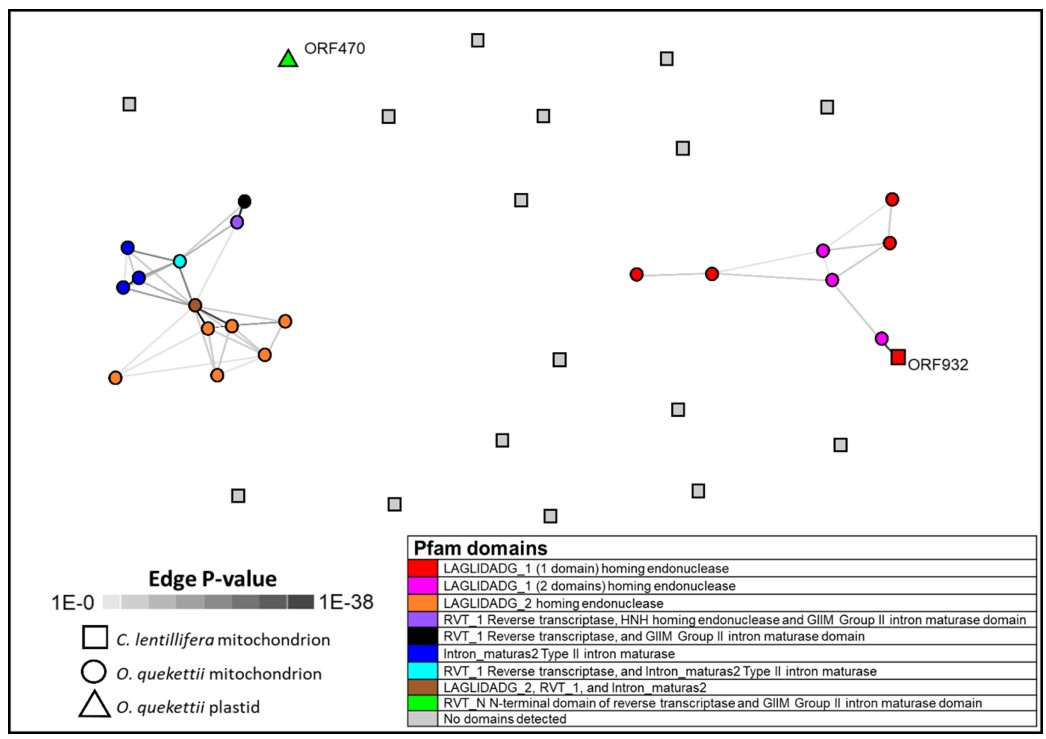

**Figure 2 Similarity network generated from all against all BLAST+ similarities of ORFs encoded in the *Caulerpa lentillifera* mitochondrion, and *Ostreobium quekettii* mitochondrion and chloroplast.** Each node represents an ORF, and each edge (line) represents a significant HSP (high scoring segment pair), shaded according to *p* value. Generated using CLANS through the MPI Bioinformatics Toolkit (Scoring Matrix BLOSSUM62, extracting BLAST HSPs up to *e*-values of 1e−4, using *p*-values better than 1.0).                                                               

mitochondrial genome, and these sequences did show convincing similarity at the DNA level in alignments, suggesting that these groups of introns proliferated within the *Ostreobium* lineage (Fig. S2).

Open reading frames with identifiable Pfam domains were only found in introns, not intergenic DNA (Table 2). Eighteen of the introns in the *O. quekettii* mitochondrial genome contain one or more ORFs. In a similarity network generated from all against all BLAST+ similarities, these ORFs appear to cluster together based on shared Pfam domains, which predict similar functions for these ORFs (Fig. 2). ORFs show homology to intron-encoded proteins that act as maturases and homing endonucleases, which enable splicing and promote intron mobility (*Lambowitz & Belfort, 1993*; *Lambowitz & Zimmerly, 2011*; *Hausner, 2012*). They contain a variety of domains, but mostly two types of domains with the amino-acid motif LAGLIDADG, which are common in both group I and group II introns (*Hausner, 2012*). Three ORFs contain double LAGLIDADG_1 domains, which in other lineages are in intron-encoded proteins with maturase activity that enable intron splicing (*Lambowitz & Belfort, 1993*). Four ORFs contain an RVT domain; *Cremen et al. (2018)* found evidence for the mobility of group II intron-encoded ORFs containing an RVT domain within Bryopsidales chloroplast genomes.

Only one ORF identified by *Zheng et al. (2018)* in *C. lentillifera* had a positive hit to the Pfam database: ORF932 in *cox*1 had a putative LAGLIDADG_1 domain. This sequence clustered with *O. quekettii* ORF sequences containing the same domain (Fig. 2).

None of the *O. quekettii* mitochondrial ORFs show similarity to the single ORF460 identified in the *O. quekettii* chloroplast, which has a putative intron splicing function (*Cremen et al., 2018*). Along with the results for the introns, this suggests there is little evidence for the transfer of ORFs or introns between organellar genomes in *O. quekettii*.

## Testing evolutionary theories: rates of evolution and mitochondrion-targeted proteins

The inflated mitochondrial genome of *O. quekettii* prompted several hypotheses (see below) that could be examined using molecular data.

In order to gain insight into whether there are differential mutation rates between the plastid and mitochondrial genomes, that might help explain their differing sizes in both *O. quekettii* and *C. lentillifera*, we aligned genes from the two organellar genomes between the two species. Estimates of base substitutions per site using a simple Jukes–Cantor model showed a slightly higher ($t = 2.94$, $p < 0.01$) number of average base substitutions per site for the mitochondrial genomes (mean $0.456 \pm 0.207$ SE) compared with chloroplast genomes (mean $0.370 \pm 0.016$ SE) (Table S5). Genes showed signs of saturated divergence, with estimated $d_S$ values considerably greater than one (e.g. *atp*6 $d_S = 46.8711$).

To identify potential homologues of nuclear-encoded DNA repair proteins that are targeted to the mitochondrion in higher plants and may play a role in organellar genome maintenance and size (see below), we performed blast searches of *Arabidopsis* sequences against the *O. quekettii* transcriptome and nuclear contigs. We identified a sequence in *O. quekettii* with strong similarity to the *Arabidopsis* DNA mismatch repair protein MSH1 and putative targeting to the mitochondrion (DeepLoc-1.0L: Mitochondrion 0.7151, Soluble 0.6145). Pfam and InterPro searches identified a putative specific DNA-binding GIY-YIG domain in this MSH1 homologue, as well as in potential homologues in other green algae. However, this domain lacks most of the key residues conserved among GIY-YIG family members (*Garrison & Arrizabalaga, 2009*). We also identified putative mitochondrion-targeted (DeepLoc-1.0L: Mitochondrion 0.9268 and 0.6334, Soluble 0.5699 and 0.6798) homologues of RECA proteins, which are also predicted to play a role in controlling mitochondrial genome maintenance in plants. Searches of the *O. quekettii* transcriptome revealed potential OSB1 homologues, predicted to be involved in homologous recombination-dependent repair, containing a central OB-fold domain but lacking a targeting signal to the mitochondrion or chloroplast (DeepLoc-1.0L: Nucleus 0.6485, Soluble 0.7227).

## DISCUSSION

The inflated mitochondrial genome of *O. quekettii*, the largest Chlorophyta mitochondrial genome sequenced thus far, even larger than that of *C. lentillifera*, lends support to large mitochondrial genomes being a characteristic of the order Bryopsidales. The *O. quekettii* mitochondrial genome encodes all genes commonly found in Chlorophyta mitochondrial

genomes, including most ribosomal protein genes, which have been unevenly retained in plant mitochondria (*Palmer et al., 2000*; *Mower, Sloan & Alverson, 2012*), and are missing in some Chlorophyta lineages such as the Chlorophyceae and *Pedinomonas* (e.g. Fig. S3). Surprisingly, although ribosomal proteins have been identified in other Ulvophyceae mitochondrial genomes thus far, none were identified in the mitochondrial genome of *C. lentillifera* (*Zheng et al., 2018*). The large size of both Bryopsidales mitochondrial genomes relative to their plastids is not typical of the Chlorophyta, where chloroplast genomes tend to be either similar size or larger and contain more noncoding DNA than their mostly compact intron-poor mitochondrial counterparts (*Leliaert et al., 2012*). Instead, these genomes are more typical of land plants: bloated with introns and intergenic DNA (*Leliaert et al., 2012*; *Mower, Sloan & Alverson, 2012*). In fact, the Bryopsidales mitochondrial genomes are larger than those of many streptophytes, above the average range cited for non-vascular land plants but not quite as extreme as has been described in the Trachaeophytes (*Mower, Sloan & Alverson, 2012*). The GC content of the Bryopsidales mitochondrial genomes is also high, markedly higher than their chloroplasts. Differing nucleotide content between organelle genomes, however, is not unheard of in green algae, nor is drastic variation in organelle nucleotide content between even closely related lineages (*Smith et al., 2011*; *Del Vasto et al., 2015*).

There is evidence for at least proliferation of group II introns within the *Ostreobium* lineage, something that is not uncommon in green algal chloroplast genomes (*Cremen et al., 2018*). However, a lack of sequence similarity for ORFs identified between their genomes, alongside the fact that their introns are not readily alignable, suggests that either the introns and associated proteins arose independently between *C. lentillifera* and *O. quekettii* before proliferating within their respective lineages, or that sequences have diverged so much that similarities are no longer recognisable. This is not unlikely given the estimated 479 million years divergence time between these lineages (*Verbruggen et al., 2009*). *Cremen et al. (2018)* did find some homologous ORFs, with conserved protein domains, between Bryopsidales chloroplast genomes. The lack of similarity between *Ostreobium* mitochondrial ORFs and its single plastid ORF with a putative intron splicing function (*Cremen et al., 2018*), as well as between introns from the organellar genomes, indicates there has been no transfer of introns between organellar genomes.

The considerable difference in their sizes and gene density as well as a lack of evidence for ORF or intron transfer between organellar compartments indicates that evolutionary forces are acting upon plastid and mitochondrial genomes differently in the Bryopsidales. The MHH (*Lynch, Koskella & Schaack, 2006*) and drift-barrier (*Lynch et al., 2016*) hypotheses emphasise the importance of mutation rate in determining genome sizes. Molecular evolution rates of the *Ostreobium* chloroplast are slow compared with other Bryopsidales; *Marcelino et al. (2016)* propose that this is due to the low light habitat of *Ostreobium*, which might reduce cases of direct sunlight-induced mutation as well as drive the evolution of slow metabolic rates and generation times with fewer mutations accumulated per unit of time. Estimates in this study indicate a slightly higher substitution rate in Bryopsidales mitochondrial genomes compared with the chloroplasts, which appears to contradict the MHH that states genomes with a higher mutation rate should

show signs of compaction, as they are under greater selective pressure to remove excess DNA that carries a mutational burden (*Lynch, Koskella & Schaack, 2006*). $d_S$ values estimated from gene alignments show signs of saturated divergence, making accurate calculation of $d_N/d_S$ unrealistic (*Yang & Bielawski, 2000*; *Hurst, 2002*). This is unsurprising due to the considerable time since the divergence of these lineages. It is not uncommon for mutation rates to vary between lineages and even between genome compartments of the same lineages by a considerably larger margin than was estimated in this study, and it can be difficult to draw direct connexions between mutation rates and genome architecture (*Smith, 2016*). An example is the green alga *Dunaliella salina*, which contains inflated organelle genomes, both chloroplast and mitochondrial, but has order-of-magnitude differences in mutation rates between the two compartments, with substitution rates between two strains of *D. salina* 2–13 times greater in mtDNA than ptDNA (*Del Vasto et al., 2015*). Differing mutation rates alone appear insufficient to explain the genome dynamics of the two Bryopsidales organellar compartments.

The higher mutation rate in the Bryopsidales mitochondrial genomes may have driven selection for more efficient DNA repair proteins, as would be predicted by the drift-barrier hypothesis (*Lynch et al., 2016*). It has been proposed that two distinct strategies have evolved to protect organelle genomes from the negative effects of non-homologous recombination. In animal mtDNA, elevated rates of evolution cause noncoding elements to pose a high mutational burden, leading to selection against noncoding elements, avoiding the build-up of repeats and introns (*Lynch, Koskella & Schaack, 2006*; *Galtier, 2011*). Plants appear to have efficient recombination-mediated DNA repair of coding DNA, explaining their low measured mutation rate (*Odahara et al., 2009*; *Davila et al., 2011*; *Christensen, 2014*). The nuclear-encoded RECA3 and MSH1 genes in plants are hypothesised to control mitochondrial genome maintenance, by preventing replication of short repeats while allowing recombination-dependent replication of longer repeats (*Shedge et al., 2007*). We identified a putative mitochondrion-targeted MSH1 homologue in *O. quekettii*. MSH1 encodes a protein with six conserved domains (*Kowalski et al., 1999*) including domain VI, a GIY-YIG homing endonuclease, which is predicted to be responsible for specific DNA-binding and suppression of homologous recombination (*Fukui et al., 2018*) and is specific to only the plant form of the protein (*Abdelnoor et al., 2006*; *Shedge et al., 2007*). Domain VI is absent from nuclear localised homologues in plants (MSH2–MSH6) and from the yeast MSH1 protein (*Abdelnoor et al., 2006*). Although InterPro and Pfam predicted a GIY-YIG domain in the *O. quekettii* MSH1 homologue, the fact that it lacks most of the residues that are typically conserved in this domain leaves its function unresolved.

*Ostreobium quekettii* also appears to encode mitochondrially-targeted RecA proteins. RecA recombinases in *Arabidopsis* are involved in strand exchange and the joining of paired DNA ends during homologous recombination (*Kühn & Gualberto, 2012*; *Gualberto et al., 2014*). RECA1 is chloroplast targeted, RECA2 is dual targeted to the mitochondrion and chloroplast, and RECA3 is targeted to the mitochondrion (*Shedge et al., 2007*). RECA3 mutations in *Arabidopsis* result in mitochondrial rearrangements similar but not identical to MSH1 mutants (*Shedge et al., 2007*). *Odahara et al. (2009)* propose that these

RecA proteins mediate homologous recombination, which is significant for suppressing short repeat-mediated genome rearrangements in plant mitochondria. They suggest that this genome stabilisation provided by RecA could allow the number of group II introns, the dominant form in the *O. quekettii* mitochondrion, to increase (*Odahara et al., 2009*). OSB1 is another putative component of homologous recombination-dependent repair in plant mitochondria, which is also likely involved in restricting mtDNA recombination (*Kühn & Gualberto, 2012*; *Gualberto et al., 2014*). Searches of the *O. quekettii* transcriptome revealed potential OSB1 homologues. However, these lack targeting signals to organelles that would provide evidence supporting their predicted function.

In contrast to the mitochondrial and chloroplast genomes of *V. carteri* (*Smith & Lee, 2009*), and those recently published for *Haematococcus lacustris* (*Zhang et al., 2019*), as well as the mitochondrial genomes of many land plants (*Palmer et al., 2000*; *Mower, Sloan & Alverson, 2012*), little of the expanded content of Bryopsidales mtDNA is repetitive DNA. However, most of the repeats in the *O. quekettii* mitochondrial genome are so-called 'intermediate' repeats (50–600 bp) (*Kühn & Gualberto, 2012*). Repeats of this length are associated with MSH1-induced recombination in *Arabidopsis* mitochondria, that can lead to accumulation of DNA as well as complex rearrangements (*Gualberto et al., 2014*). Along with error prone repair these processes might result in low numbers of alternative genome configurations, 'mitotypes' that could eventually increase to become the dominant form of mtDNA (*Gualberto et al., 2014*). This working hypothesis, based on molecular data, proposes that a recombination-associated repair process in the Bryopsidales mitochondrion has resulted in its inflated size, and encourages further study to resolve the role played by these putative mitochondrion-targeted sequences as well as to determine whether recombination is in fact occurring in Bryopsidales mitochondrial genomes at all.

It is interesting to consider whether the build-up of introns in Bryopsidales mitochondrial genomes plays a part in modifying recombination if it is in fact occurring. Introns may play a role in reducing intragenic Hill–Roberston effects, that tend to result in locally reduced $N_e$, by increasing recombination rate between regions of the genome (*Comeron & Kreitman, 2000*). Bottlenecking of genomes, such as during the production of gametes for sexual reproduction, can also result in smaller genome $N_e$ (*Neiman & Taylor, 2009*), which would reduce selection and might contribute to the higher mutational load observed in Bryopsidales mitochondrial genomes. Sexual reproduction has not been observed in *O. quekettii*, however it has been described in other Bryopsidales (*Morabito, Gargiulo & Genovese, 2010*) and, to our knowledge, no strictly asexual lineages have been observed. Comparing relative copy numbers of organelle genomes between adult plants and gametes could provide insight into the extent of bottlenecking that occurs during reproduction in the Bryopsidales.

The mutation and recombination rates in organelles are at least in part under the control of nuclear-encoded maintenance pathways (*Smith & Keeling, 2015*), with organelle genomes usually lacking the genes necessary for their own DNA replication and repair (*Sloan & Taylor, 2012*). It has been proposed that the evolution of similar features in both plastid and mitochondrial genomes within a species, as seen in some of the Chlorophyta, is due to 'leakage' of nuclear-encoded proteins controlling these processes between

organellar compartments, with proteins normally targeted to one organelle also becoming targeted to the other (*Smith & Keeling, 2015*). The differing configurations of the two organellar genomes in *C. lentillifera* and *O. quekettii* suggest that this has not occurred in the Bryopsidales. As more nuclear genomes for Bryopsidales become available (*Arimoto et al., 2019*), there will soon be the opportunity for a more thorough study of all three genomic compartments in the Bryopsidales, including an examination of organelle-targeted DNA maintenance machinery to further uncover the forces underpinning their divergent organelle genome sizes.

## CONCLUSIONS

*Marcelino et al. (2016)* suggested that selection due to a low light environment resulted in the reduced chloroplast genome of *O. quekettii*, but it is unclear why this selection would not act against the expanded genome size in the mitochondrion. It would appear that effective population size, recombination and/or mutation rate, influenced by nuclear-encoded proteins, are different between the two genomes, leading to a reduction in the strength of selection to influence evolution of the mitochondrial genome. Ultimately, it is likely overly simplistic to attempt to find a single explanation to cover all mitochondrial genome expansion (*Smith & Keeling, 2015*), with the evolution of organellar genomes in the Bryopsidales and other lineages a combination of many forces and factors.

## ACKNOWLEDGEMENTS

This research was supported by use of the computational facilities of the University of Melbourne High Performance Computing platform and the Nectar Research Cloud. This paper benefitted from useful discussions with Joseph Bielawski about evolution, as well as helpful comments from Geoffrey McFadden and Patrick Buerger on an earlier draft. We appreciate the thoughtful comments on the manuscript made by Jiri Neustupa and an anonymous reviewer.

### Funding

Funding was provided by the Australian Research Council (DP150100705) and the University of Melbourne (Early Career Researcher grant to Heroen Verbruggen and Computational Biology Research Initiative seed funding to Heroen Verbruggen, Kathryn Holt and Linda Blackall). The funders had no role in study design, data collection and analysis, decision to publish, or preparation of the manuscript.

### Grant Disclosures

The following grant information was disclosed by the authors:
Australian Research Council: DP150100705.
University of Melbourne.

### Competing Interests

The authors declare that they have no competing interests.

## Author Contributions

- Sonja I. Repetti conceived and designed the experiments, performed the experiments, analysed the data, prepared figures and/or tables, authored or reviewed drafts of the paper, approved the final draft.
- Christopher J. Jackson conceived and designed the experiments, performed the experiments, analysed the data, authored or reviewed drafts of the paper, approved the final draft.
- Louise M. Judd performed the experiments, authored or reviewed drafts of the paper, approved the final draft.
- Ryan R. Wick performed the experiments, authored or reviewed drafts of the paper, approved the final draft.
- Kathryn E. Holt conceived and designed the experiments, contributed reagents/ materials/analysis tools, authored or reviewed drafts of the paper, approved the final draft.
- Heroen Verbruggen conceived and designed the experiments, contributed reagents/ materials/analysis tools, authored or reviewed drafts of the paper, approved the final draft.

## DNA Deposition

The following information was supplied regarding the deposition of DNA sequences:

The *O. quekettii* mitochondrial genome sequence is available at GenBank: MN514984.

## Data Availability

MAFFT alignments of ORF-lacking intron clusters identified in Fig. S2 are available in a Supplemental File.

Raw RNA sequencing data is available at ENA: PRJEB35267.

## Supplemental Information

Supplemental information for this article can be found online at http://dx.doi.org/10.7717/peerj.8273#supplemental-information.

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
