# Peer review of "The inflated mitochondrial genomes of siphonous green algae reflect processes driving expansion of noncoding DNA and proliferation of introns"

_PeerJ, doi:10.7717/peerj.8273_

## Round 0.1 · original submission · Minor Revisions

The two reviewers agree that this is a worthy study, but both make valid, specific suggestions to improve the manuscript by sharpening the message and clarifying the data interpretation and refining the wider context of the study in the Discussion. Please consider these concerns carefully in a revised manuscript.

·

Basic reporting

This is a well prepared and interesting manuscript. I am sure that it deserves to published and it would certainly be a useful addition to curent knowledge of algal genome evolution.

Experimental design

no comment

Validity of the findings

The key finding seems to be that in Ostreobium the compressed chloroplast genome is not accompanied by a similarly compressed mitochondrial genome. Apparently, this limits plausibility of the "low-light selection" hypothesis that was used as the likely explanation of the plastid genome patterns in this sciaphilic organism. The authors provide a number of alternative explanations; however, given the very limited number of available bryopsidalean mitochondrial genomes (just 2), the explanatory power of their interpretations is necessarily limited. It might be useful to expand the scope of the "Discussion" to other chlorophytan green algae with available mitochondrial and plastid genome data, such as other members of Ulvophyceae and even other UTC-clade lineages. This is also suggested in individual points to particular statements and paragrpahs of the text (see below).

Additional comments

In general, I think that the manuscript should be accepted for the journal after a minor revision and, possibly, addition of a more general context ivolving other Chlorophyta into the discussion of the data. This might make an even more interesting and probably more frequently cited publication.

1) It should be mentioned in the "Abstract" that only one of the studied mitochondrial genomes has actually been newly published as part of the study (Ostreobium quekettii).

2) l. 59: "approximately 60% noncoding DNA in smaller lineages"
It is unclear what you mean by "smaller lineages". Smaller by relatively lower number of described taxa, available genome data, or even smaller body sizes? Subsequent sentences that discuss genomes of Volvox carteri keep open all these possibilities.

3) l. 63: "the large non-coding DNA content in V. carteri genomes supports the mutational-hazard hypothesis (MHH)"
The MHH hypothesis presumes that non-coding DNA accumulates in taxa with slow mutation rates (e.g., slow reproduction modes) and low population numbers. However, what data do you have to support your notion that V. carteri has smaller population numbers (or reproduction modes) than other phototrophic eukaryotes? The population numbers do not depend just on the number of cells in colonies but also on niche breadth, distribution area, etc. In my view, your statement is a pure speculation and it should be stated as such.


4) l. 70-71: "There is expected to be a large reduction in..." < "A large reduction in ... ... is expected."

5) l. 190: ENA #

6) l. 285-286: "the largest green algal mitochondrial genome sequenced thus far, even larger than that of C. lentillifera."
This would perhaps better fit into Discussion.
In addition, it might be interesting to compare the size of the genome with streptophytan lineages, too.

7) l. 295-301: The GC content
How usual is it that the GC content of the mitochondrial genome vastly exceeds the GC content of the plastid genome of the same organism? It might be useful to menation a few examples.

8) l. 383-384: "ribosomal protein genes ... that surprisingly appear to be missing from the mitochondrial genome of C. lentillifera"
What about in other members of the UTC clade with known mitochondrial genome data?

9) l. 484: "Sexual reproduction has not been observed in O. quekettii..."
It would be quite surprising if it turned out to be a truly asexual genus/species. Are there any known members of Bryopsidales with reasonable evidence of purely asexual way of life?

10) l. 494-495: "The differing configurations of the two organellar genomes in Caulerpa lentillifera and Ostreobium quekettii..."
I understand that with just two members of the Bryopsidales any interpretation of different patterns of bryopsidalean plastid and mitochondrial genomes is limited. However, maybe you could provide more comparison with other members of the Ulvophyceae and even the UTC clade. Of course, these algae are even more distant but the estimated divergence time of Ostreobium and Caulerpa in early Ordovician makes eventual correlated patterns between the two taxa less probable, anyway.

Reviewer 2 ·

Basic reporting

The article is definitely interesting and worthy of publishing - it presents new data, thoughtfully and thoroughly analyzed; it is well written and pitched to a broad audience with interest ranging from algal genomes to evolutionary theory.

The article is appropriately structured and the data are disclosed - it appears that the genbank number MN514984 has not yet been released, so that should be done before the article is published. I also ask in my general comments if the authors could submit their intron alignment in a more usable format - like plain text or fasta.

Experimental design

On the whole, the manuscript's methods are sound and appropriately described. I only have one comment regarding one particular part of the methods:

Assessing homology in highly variable sequences like introns is quite tricky. So, I generally view any alignment and phylogeny of introns with a level of skepticism. I appreciate that the authors acknowledged that in most cases the homology was not demonstrable and therefore the data not phylogenetically analyzable. However, more information on the analysis could be useful. Were only the conserved regions of the introns analyzed? Or their entire sequences? Focusing the analysis on a small, yet 'cleaner' subset of the data might have been insightful.

Validity of the findings

All conclusions are justified, and even though much of the evolutionary interpretation comes with great uncertainty, the authors adequately discuss the possible explanations and applicable hypotheses described in previous literature.

Additional comments

I have a few specific comments on the text, figures and data that I think could use some clarification, but they are all rather minor:

L54 - The term "protist" is rather nebulous, and I think it would be better to be specific. So, not just in green algae, but also in other photosynthetic (or also non-photosynthetic?) eukaryotes? Evolutionarily the term protist doesn't have much meaning, so I think in an evolutionary-slanted paper it would be better to be precise.

L90 and beyond
Some of the pieces of evolutionary theory are explained very clearly and extensively, so it is surprising that hitchhiking and background selection are terms just tossed in with the rest without further explanation. I realize that these are all related to the same topic/phenomenon, but not every reader might know this connection right away - so I think as the rest of the theory is well explained, these terms should be elaborated on adequately too.
Further on in the same paragraph, if I understand it correctly, it says that regions with lower recombination rates have more and bigger introns - but then a later statement says that introns tend to increase recombination rates. These two seem contradictory to me, even though I think I understand how both can be true. Does it mean that low-recombination regions tend to get invaded by introns more, but in turn those introns themselves can then serve as recombination sites?
It would be good to have a bit more clarification on how these two statements are connected.

L106 - "therefore differing Ne" - this could use a more explicit explanation. It is explained above how multicellularity and Ne are connected, but not complexity or diversity of cell/tissue types.

At times, in the discussion especially, it gets a bit murky whether the authors are talking about populations (and population sizes) of the organisms, or populations of the organelles and their genomes. Given that there can be numerous organelles per cell, and they may not always be identical genetically, this could use some extra specificity in the text.

Maybe this is just the review pdf, but Fig. 1 is not of very good quality, quite fuzzy.
I like Fig.1, it is very informative, but I'm not sure how length of repeats is shown in the innermost circle - it doesn't seem discernible. Is it the shade of the lines that represents the length?

In Fig 2 the one LAGLIDADG2 reverse transcriptase being so far from other LAGLIDADGs - does this mean that the LAGLIDADGs are not monophyletic? or that the one (brown square) has acquired a different domain in addition to the LAGLIDADG motif? I find this quite intriguing; maybe a bit more explanation would be good in the text.

Supplements: Alignments in Fig S2 are appreciated, but not very usable in pdf form. Could the authors also submit a plain-text version of these?

The supplementary tables (especially those that are in word docs) should have some sort of captions too, so that the reader doesn't have to look up what they represent.

---

## Round 0.2 · accepted · Accept

The authors have carefully addressed all issues raised by the reviewers.